Hybrid deep layered network model based on multi-scale feature extraction and deep feature optimization for acute lymphoblastic leukemia anomaly detection

Çınarer Gökalp gokalp.cinarer@bozok.edu.tr
Computer Engineering, Yozgat Bozok University , Yozgat , Turkey
Coelho Paulo Jorge
Electronic publication date: 2025 Sep 4
Publication date: 2025
Volume: 11
Electronic Location ID: e3174
Received 2025 Apr 17; Accepted 2025 Aug 7
Copyright: © 2025 Çınarer
Copyright year: 2025
Copyright holder: Çınarer
License: This is an open access article distributed under the terms of the Creative Commons Attribution License, which permits unrestricted use, distribution, reproduction and adaptation in any medium and for any purpose provided that it is properly attributed. For attribution, the original author(s), title, publication source (PeerJ Computer Science) and either DOI or URL of the article must be cited.
License URL: https://creativecommons.org/licenses/by/4.0/

Keywords: Deep learning, Machine learning, ALL detection, Convolutional neural network (CNN), Transfer learning, Hybrid deep learning

Funding: The authors received no funding for this work.

==============================
Acute lymphoblastic leukemia (ALL), one of the common diseases of our day, is one of the most common hematological malignant diseases in childhood. Early diagnosis of ALL, which plays a critical role in medical diagnosis processes, is of great importance especially for the effective management of the treatment process of cancer patients. Therefore, ALL cells must be detected and classified correctly. Traditional methods used today prolong the detection and classification processes of cells, cause hematologists to interpret them according to their expertise, and delay medical decision-making processes. In this study, the performance of the hybrid model developed with different deep learning models for ALL diagnosis was comparatively analyzed. In the proposed ALL detection architecture, blood cell images were processed using the center-based cropping strategy and irrelevant areas in the images were automatically removed. The dataset was divided into training, validation, and test sets, and then features were extracted with deep hyperparameters for convolution, pooling, and activation layers using a model based on Xception architecture. The obtained features were optimized to the advanced Extreme Gradient Boosting (XGBoost) classifier and model classification results were obtained. The results showed that the proposed model achieved 98.88% accuracy. This high accuracy rate was compared with different hybrid models and it was seen that the model was more successful in detecting ALL disease compared to existing studies.

Introduction

Leukemia is one of the most common types of blood cancer today. This type of cancer occurs due to abnormal increases in the production of immature blood cells in the bone marrow. In the United States, 20,700 new cases of chronic lymphocytic leukemia are estimated among all new cancer cases in 2024, while an estimated 215,107 people will be living with chronic lymphocytic leukemia (Hallek, 2025). The rapid proliferation of abnormal white blood cells (WBCs) affects the blood and bone marrow. These abnormal cells cannot resist diseases and negatively affect the bone marrow’s ability to produce red blood cells and platelets (Almadhor et al., 2022). Leukemia can be seen in two different types: acute or chronic. Acute leukemia develops more quickly than chronic leukemia and is described as the most aggressive type, as it shows more intense symptoms.

ALL is caused by the accumulation of a large number of lymphoblasts in the bone marrow and peripheral blood (de Sant’Anna, de Oliveira & Dantas, 2022). While ALL is mostly diagnosed in children (80%), it is known to be fatal when it is seen in adults. The incidence of ALL in the USA is estimated to be 1.6 per 100,000 people (Terwilliger & Abdul-Hay, 2017). In general, doctors suspect ALL patients based on certain symptoms and findings, while different clinical examinations confirm this diagnosis. In the early stages, blood samples are often taken from patients with suspected ALL and tests are performed. In these tests, complete blood count and peripheral blood smear examinations are performed to monitor changes in the number and appearance of white blood cells and normal blood cells, respectively. Different colors can be seen in each lot in microscopic images (Gupta et al., 2020).

ALL is diagnosed with a higher accuracy rate using chromosome-based tests. Early diagnosis is of great importance in this disease. With early diagnosis, the survival rate can be up to 90%. Therefore, it is a great necessity to diagnose ALL patients at earlier stages and start treatment earlier (Mondal et al., 2021). In this context, artificial intelligence (AI) algorithms and applied deep learning (DL) models play a very important role. AI has attracted great attention in many areas in recent years (Tai, 2020). Artificial intelligence technologies are used in science, health, industry and many sectors (Davenport & Kalakota, 2019). The various advantages provided by artificial intelligence also allow us to achieve serious gains in the field of cell biology (Fu, 2019). Hematological diseases include a series of pathological conditions that occur due to numerical and functional disorders of cells in the blood, and are a serious health problem worldwide and constitute the majority of diseases such as cancer (Castillo et al., 2012).

Blood cancer is diagnosed by microscopic examination of blood samples, which is one of the traditional methods. Traditional methods for detecting blood cancer include complete blood count, peripheral smear, bone marrow aspiration and biopsy, immunophenotyping, cytogenetic and molecular tests, biochemical analyses, and imaging methods when necessary. Traditional tests are generally time-consuming in terms of laboratory procedures, expert examinations, and interpretation of results. Traditional methods only evaluate limited data that can be seen with the eye or measured with a test; deep learning can detect deeper patterns in complex data. However, it may be difficult to distinguish leukemic blasts from normal cells under the microscope, because the images of the two cells may be similar (Liu et al., 2015). At the same time, the application of traditional methods with manual and intuitive approaches may delay the determination of possible risks and the determination of the responses to be developed accordingly. Classical ML and DL algorithms are insufficient to eliminate unnecessary information in the feature map generation process, and overfitting cannot be prevented in convolutional neural network (CNN)-based models (Jawahar et al., 2024). Intuitive approaches and models created based on these approaches provide lower precision in risk assessment in image analysis. In addition, the most basic disadvantages of manual procedures used in image analysis include loss of time and heavy workload of pathologists. The hybrid model we proposed overcomes these difficulties by using features obtained from deep neural networks, allowing us to extract more discriminative features with fewer trained parameters. The main purpose of this research is to demonstrate the accuracy and reliability offered by the proposed hybrid model in detecting blood cells, unlike deep learning models.

The contributions in this study are summarized as follows:

Inductive parameter transfer hybrid learning approach is proposed for ALL prediction.

It is shown that the architecture of the proposed model, preprocessing steps and the applied model are the most successful method for the diagnosis of ALL images.

The model is extensively tested with different deep learning algorithms to determine the effects of the proposed model.

DenseNet201, InceptionV3, Xception, VGG19, ResNet50 are examined as basic architectures with layer-wise and block-wise hyperparameter settings to learn the feature representation of ALL images and their classification performances are comparatively analyzed.

The determination of the most suitable optimizers for ALL detection and the examination of their performance metrics are provided.

In order to improve the prediction performance, datasets are subjected to preprocessing such as thresholding, data augmentation and tested with hybrid models and comparative performance analysis is performed.

This article demonstrates the high-performance success of the proposed custom-tuned CNN-based hybrid model. The workflow of the proposed method is illustrated in Fig. 1.

Figure 1 The workflow of the proposed method.

Literature review

A compilation of algorithms developed by studies on automatic diagnosis and treatment of hematological diseases using the C-NMC dataset is presented. Goswami et al. (2020) developed a model using the C-NMC 2019 challenge dataset. Due to the small number of subjects in the dataset, the researchers divided the training set into seven layers proportionally and trained each layer separately with seven different CNN networks. The researchers used the Inception-v3 deep CNN as a model. For better training, the authors used various techniques such as Horizontalflip, Verticalflip, RandomRotation, Affine Transformation, tone adjustment, fullness, contrast, brightness and RandomCrop, which resizes the images to 299 × 299 for Inception-v3. As a result of all these operations, the performance of the model is 95.24% as a weighted F1-score. In another study, a deep belief network model was proposed for the segmentation of white blood characteristics developed from acute ALL microscopic images (Duggal et al., 2016).

Ramaneswaran et al. (2021) combined the deep learning model Inception v3 for image features and the XGBoost algorithm for clinical features. This hybrid model aims to make a more accurate classification by combining the features obtained from leukemia cell images and clinical data. The results showed that the hybrid model provided higher classification accuracy compared to models made using only Inception v3 or XGBoost. The researchers also compared previously trained single models and determined that Inception v3 gave the best result among them. Resizing, center cropping, data augmentation and normalization were performed in data preprocessing. Inception v3 model performed with 0.979 accuracy rate, while the hybrid model performed with 0.985 accuracy rate. There are also hybrid models in the literature with different datasets. Kasani, Park & Jang (2020) presented ensemble deep learning models for the classification of leukemia B-lymphoblasts using the SBILab dataset. The researchers normalized the data using different data preprocessing techniques. The ensemble models used were NASNetLarge + VGG19, which gave the highest accuracy rate of 96.58%.

Ghaderzadeh et al. (2022) presented a model for automatic detection of normal B-lymphoid precursors and leukemia B-lymphoblasts from blood smear microscopic images. This model was developed using image processing techniques and artificial neural network (ANN) algorithms. As a result of the preprocessing applied in the study, different morphological features and cell data were obtained using feature extraction techniques. These features were then combined as a feature vector to be used in the ANN model and the model was trained. In the test set, the performance of the model was evaluated with architectures consisting of Xception, ResNet-v2, İnception-v3 and DenseNet1212. The highest F1-score obtained by the model is 98.4%

Sampathila et al. (2022) presented a customized deep learning classifier using blood smear images for the diagnosis of ALL. In the study, the “ALL Challenge Dataset of ISBI, 2019” dataset was used (Sampathila et al., 2022). The first step of the study is to prepare the image dataset. At this stage, the dimensions of the images were arranged and the normalization process was applied. In the second step, various filtering techniques were used to reduce noise in the images and make the cells more distinct. In the third step, the model was designed and trained. In this process, a classifier model was developed using CNN. For training the model, the dataset was divided into training, validation and test sets, and hyperparameter optimization was performed. The developed model was used to classify ALL cells and normal cells and an accuracy rate of 95.54% was obtained.

Mathur et al. (2020) proposed a customized deep learning classification model called mixup multi-attention multi-tasking model (MMT) for the diagnosis of early stage leukemia. The ISBI 2019 CNMC data set was used for this model. Data preprocessing operations include data normalization, sizing, and histogram equalization methods. Then, the dataset was augmented using the Mixup method by combining images illuminated from different angles. Performance measurements were made by comparing this model with other models. The F1-score rate of the model is 91.89% (Mathur et al., 2020).

In another study, Almadhor et al. (2022) propose an ensemble automatic prediction approach using the C-NMC leukemia dataset with the ensemble Voting model and four machine learning algorithms: K-nearest neighbor (KNN), support vector machine (SVM), random forest (RF), and naive Bayes (NB). The Voting model includes RF, KNN, and SVM algorithms. First, the data was preprocessed. In this study, the endpoint calculation methodology, thresholding, contour calculation, and resize operations were applied for data preprocessing. Pre-trained CNN based deep neural network architectures (VGG19, ResNet50, or ResNet101) were used for feature extraction. Three techniques were applied to select the obtained features: ANOVA, recursive feature elimination (RFE), and RF. In the study, the SVM algorithm achieved the highest accuracy rate of 90.0% (Almadhor et al., 2022).

Tusar & Anik (2022) proposed an automatic system to detect ALL cells using ALL image dataset prepared in Taleqani Hospital Bone Marrow Laboratory (Tehran, Iran) and deep neural networks (DNN). The researchers used resize, padding, augmentation and normalization methods in data preprocessing. The DNN algorithms used are Convolutional Neural Network (ConvNet), MobileNetV2, Residual Neural Network 50 (ResNet50), Visual Geometry Group 2019 (VGG19). Among these models, MobileNetv2 achieved the highest accuracy rate with 0.9742 (Tusar & Anik, 2022).

Mondal et al. (2021) discussed a CNN architecture used to diagnose ALL from microscopic images using the C-NMC-2019 dataset (Mondal et al., 2021). In this study, different CNN architectures were combined to improve the performance of the ensemble model. These models are VGG16, InceptionV3, ResNet50, Xception and DenseNet121. The authors used two different methods to improve the performance of the ensemble model. The first method is to combine the predictions of five different CNN models, get the average predictions and determine a threshold value to classify the results. The second method is to create a new feature set by combining the outputs of five different models and perform the next classification stage using this feature set. Augmentation, rebalancing, ROI extraction methods were applied for preprocessing the data.

Alharbi et al. (2022) proposed the UNet algorithm for segmentation and classification of white blood cells using the ISBI C-NMC 2019 dataset. The features obtained from the images were processed using feature extraction methods and then the UNet model was trained for segmentation and classification of the features.

Deshpande, Gite & Aluvalu (2021) conducted a study investigating different areas where microscopic imaging of blood cells is used for disease detection. Microscopic analysis using image processing, computer vision and machine learning is the main focus of the analysis and review. In this study, many datasets (Acute Lymphoblastic Leukemia Database (ALL-IDB), Atlas of hematology by Nivaldo Mediros, ASH image bank, Leukocyte Images for Segmentation and Classification (LISC), C-NMC dataset) and algorithms were investigated. As a result, it was concluded that different performance values and parameter settings are within the scope of improvement. There are three main features in this architecture, namely feature reconstruction, feature extraction and feature fusion. The accuracy rate of the proposed model was found to be 94.14%. It is seen that the approaches and models proposed in the literature reflect the general classical architectures and the researchers are not supported by new approaches. Apart from this, the literature review consisting of current publications is also included in the study. In this context, some studies conducted in the literature with similar datasets and the methods used in their implementation are given in Table 1.

Table 1 Literature review.

Reference	Method	Accuracy (%)	Precision (%)	F1-score (%)	Recall (%)	Description	
Maurício de Oliveira & Dantas (2021)	VGG16	92.48	91.14	92.60	94.10	Data augmentation was performed.	
VGG19	91.59	90.06	91.75	93.51	
Xception	90.41	87.64	90.76	94.10	
Srinivasan et al. (2021)	Inception v3	0.979	0.979	0.979	0.979	The models were analyzed as single and hybrid approaches.	
AlexNet	0.894	0.901	0.889	0.894	
DenseNet 121	0.869	0.876	0.871	0.869	
ResNet 18	0.917	0.919	0.917	0.917	
VGG 16	0.924	0.927	0.924	0.924	
SqueezeNet	0.932	0.936	0.932	0.932	
MobileNet v2	0.958	0.958	0.958	0.958	
Hybrid Model	0.985	0.972	0.986	0.976	
Duggal et al. (2017)	AlexNet	87.90	–	88.12	–	Using the T-CNN architecture prefixed with the proposed SD-Layer.	
T-CNN	92.48	–	92.70	–	
AlexNet + SD-Layer	88.50	–	88.32	–	
T-CNN + SD-Layer	93.20	–	93.08	–	
Kasani, Park & Jang (2020)	NASNetLarge + VGG19	96.58	96.94	94.67	91.75	Data augmentation was performed.	
DenseNet201 + InceptionV3	94.73	96.11	91.43	87.18	
DenseNet201 + VGG19	95.55	92.43	93.16	93.91	
InceptionV3 + VGG19	93.80	93.45	90.03	86.86	
DenseNet201 + VGG19 + InceptionV3	95.45	95.30	92.76	90.38	
Liu et al. (2022)	VGG-16	94.78	94.99	94.76	94.76	Four different datasets was used.	
ResNet-34	95.56	95.70	95.56	95.57	
ResNet-50	96.70	96.77	96.69	96.68	
GoogLeNet	95.00	95.34	94.93	94.96	
Inception-v3	96.51	96.66	96.52	96.52	
ResNeXt-50-32×4d	96.83	96.94	96.82	96.81	
ShuffleNet-v2×1	94.45	94.66	94.44	94.47	
MobileNet-v2	95.72	95.96	95.71	95.73	
MobileNet-v3-Large	95.36	95.67	95.34	95.37	
EfficientNet-B0	95.81	95.95	95.80	95.81	
Ghaderzadeh et al. (2022)	DenseNet 121	85.3	70.9	x	98.5	Feature extraction, pixel-level global mean, and standard deviation (STD) were calculated for all images.	
Inception-V3	89.77	72.4	x	99	
AlexNet	78.6	69	x	88	
ResNet-50	74	66	x	79	
ResNet-101	79	68	x	81	
Inception-ResNet-v2	89.1	68.83	x	99.92	
SqueezeNet	81	68.8	x	89	
Xception	94.91	85.65	x	99.84	
Ensemble model	x	99.4	98.4	96.7	
Sampathila et al. (2022)	ALLNet	95.54	96	95.43	95.91	Thresholding was applied. Data augmentation was performed.	
Mathur et al. (2020)	MMA-MTL	X	0.8211	0.9189	0.9385	Data preprocessing includes data normalization, resizing, and histogram equalization methods.	
ResNet-18	X	0.6568	0.8056	0.8493	
Efficient-Net	X	0.6574	0.8473	0.9045	
ResNet-18 + GAC	X	0.6164	0.8762	0.9311	
ResNet-18 + LAC	X	0.7360	0.8767	0.9389	
ResNet-18 + Mixup	X	0.7515	0.8605	0.8344	
Almadhor et al. (2022)	KNN	83.4	84.1	88.5	93.3	The endpoint computation methodology involved applying processes such as thresholding, contour calculation, resizing, and feature selection.	
SVM	90.0	90.2	92.9	95.7	
RF	82.6	82.1	88.2	95.3	
NB	76.7	85.2	82.4	79.8	
Voting	87.4	86.5	91.3	96.6	
Tusar & Anik (2022)	MobileNetV2	0.9742	x	x	x	Resize, padding, augmentation, and normalization methods were used in data preprocessing.	
VGG19	0.9613	x	x	x	
ConvNet	0.9128	x	x	x	
ResNet50	0.8526	x	x	x	
Mondal et al. (2021)	VGG-16	0.851	0.849	0.849	0.851	Augmentation, rebalancing, and ROI extraction methods were applied for data preprocessing.	
Xception	0.859	0.865	0.860	0.859	
MobileNet	0.843	0.845	0.844	0.843	
InceptionResNet-V2	0.837	0.839	0.838	0.837	
DenseNet-121	0.829	0.827	0.826	0.829	
Ensemble Model	0.898	0.897	0.897	0.898	
Vinnarasi et al. (2024)	Traditional CNN	91.65	92.5	91.75	91.5	In the proposed model, the VGG-16 Net model was used as the feature extractor, and Softmax was used as the classifier.	
AlexNet	94.6	94.5	94.5	94.5	
VGG-16	97.44	97.5	97.5	97.5	
Alharbi et al. (2022)	UNet	93.4	92.55	94.50	97.12	To maintain data integrity, standardization and normalization were applied.	
SegNet	92.14	98.77	99.10	97.66	
FCN	91.34	95.65	98.67	96.77	
Proposed method	94.14	98.45	98.67	97.56	

Materials and method

Dataset

In order to automatically analyze cancer cells of ALL patients, a competition was organized in ISBI 2019. In this competition, a dataset named ALL Challenge dataset of ISBI 2019 (C-NMC 2019) was created (Mourya et al., 2019; https://doi.org/10.7937/tcia.2019.dc64i46r). This dataset was shared publicly as C-NMC 2019 B-ALL classification challenge in The Cancer Imaging Archive (TCIA) as a dataset with detailed descriptions of training and testing parts of the subjects (Gupta et al., 2020). This dataset was evaluated in three stages. This set contains a total of 2,586 cell images and was segmented by expert oncologists (Gupta et al., 2020; Duggal et al., 2017, 2016). The training set consists of 73 subjects and consists of 47 images with leukemia and 26 images with normal cells, and this set contains a total of 10,661 cell images. The first pre-test set consists of 28 subjects and consists of 13 images with leukemia and 15 images with normal cells. This set contains a total of 1,867 cell images. Finally, the final test set consists of 17 subjects, nine of which have leukemic cells and eight of which have normal cells. The images of the dataset are shown in Fig. 2.

Figure 2 Images from the dataset.

In this study, the training set of the CNMC-2019 dataset was used. This set was divided into 70% training, 15% validation and 15% test. The total number of images in the training set is 7,462, the total number of images in the validation set is 1,599 and the total number of images in the test set is 1,600. The number of images used in the experiments is presented in Table 2.

Table 2 Dataset details.

	Train	Validation	Test	
ALL	5,090	1,081	1,091	
Normal	2,372	508	509	
Total	7,462	1,599	1,600	

Deep transfer learning models

Deep transfer learning architectures allow us to use the weights of models previously trained on large amounts of images in different tasks in a new classification task. This allows the model to be trained faster and to reach the optimum result faster. At the same time, transfer learning architectures eliminate the disadvantage of training the model from scratch on a small amount of images, allowing high accuracy performance to be achieved.

ResNet 50 model

Residual network (ResNet) is a deep learning model developed by Microsoft researchers in 2015. The aim of ResNet is to solve the problem of performance degradation that occurs as networks become deeper. This performance degradation is related to the backward propagation of the network in deeper layers and is called “degradation” (He et al., 2016). ResNet uses an innovative connection technique such as “residual connections” to solve the degradation problem. In this technique, the outputs of the layers are directly added to the input of the next layer, ensuring that the features of the previous layers are preserved. The model consists of 50 layers and has a more complex structure using “bottle-neck” layers. Bottle-neck layers are used to obtain a lighter and faster model. ResNet-50 was trained on the ImageNet dataset.

Xception model

Xception is a deep learning model called “Extreme Inception”. Xception was developed by Google and is built on the CNN architecture with 36 convolutional layers (Chollet, 2017). Xception model accepts red-green-blue (RGB) images as input and scales its size to 299 × 299. The model first creates feature maps using several convolutional layers. Then, these feature maps are processed with a type of convolutional layer called “depthwise separable convolution”. Depthwise separable convolution processes feature maps more lightly, allowing the model to be faster and with fewer parameters. This process is applied separately for each pixel in each feature map. In models such as SqueezeNet, the sampling is placed at the end of the network, allowing the convolution layers to have large activation maps (Iandola et al., 2016).

The Xception model then adds several more depthwise separable convolutional layers and finally creates a classifier with fully connected layers. This classifier is used in tasks such as object recognition, classification and detection. Finally, this model is trained on the ImageNet dataset.

VGG19 model

The VGG19 model accepts RGB images as input and scales its size to 224 × 224 (Simonyan & Zisserman, 2014). Its first layer processes the input image using 64 different filters and obtains a 64-channel image as output. Then, it adds four more blocks, each with two convolutional layers. Each block further processes the feature maps and reduces their size to obtain higher-level features. Then, it adds three fully connected layers and finally passes to the softmax classifier. This classifier is used in tasks such as object recognition and classification. In the VGG19 model, the weights of each layer are optimized with the training data. The training data is a large image dataset consisting of 1,000 categories, usually called the ImageNet dataset (Bansal et al., 2023).

EfficientNetB3 model

Efficient-Net is a deep learning model that requires less computational power to have higher accuracy rates. It was developed by Google Brain (Tan & Le, 2019).

Efficient-Net is a scalable deep learning model and regularly scales the feature maps to obtain better feature extraction by using the features at each level of the network in the best way. Efficient-Net scales the model size and depth using a method called “Compound Scaling” (Kumar & Gunasundari, 2024). This method combines the width, depth and resolution of the network to create a scalable model. It also uses an activation function called "swish". Swish is a nonlinear function like ReLU, but provides a smoother transition.

DenseNet-121 model

DenseNet-121 is a CNN model developed in 2017 (Huang et al., 2017). DenseNet is known for features called “dense connections”. These connections are created by using the output of each layer as the input of the next layer. In this way, the features of the previous layers are also used by the new layers. DenseNet-121 is a 121-layer model. It accepts 224 × 224 RGB images as input. The model consists of several dense blocks, normalization layers and pooling layers. Dense blocks are a set of layers where dense connections are used, and each block transforms the features of the previous layers into new features. Normalization layers are used to prevent the overfitting problem. Pooling layers reduce the size of the feature maps and provide local translation invariance of the features (Arathi & Dulhare, 2023). DenseNet-121 also uses bottleneck layers. These layers help in learning smaller feature maps and reduce the number of parameters of the model.

MobileNetV2 model

MobileNet V2 is an artificial neural network architecture developed by Google and is designed specifically for use in applications used on mobile devices. MobileNet V2 is designed to provide better performance, lower error rates, faster operation, and less memory consumption than its previous version (Sandler et al., 2018). MobileNet V2 can learn the weights and parameters of the network better by using a technique called kernel learning. It also uses a structure called “depthwise separable convolution”.

InceptionResNetV2 model

The InceptionResNetV2 model is a deep neural network model that uses both Inception and ResNet architectures together. It performs maximum pooling operations with convolutions of 1 × 1, 3 × 3, 5 × 5 dimensions (Rashid et al., 2024). It continues the operations by adding the features learned in each layer to the next layer to prevent gradient loss. Dimension reduction is performed with reduction blocks. Global average pooling (GAP) is applied in the last step.

InceptionV3 model

Inception V3 is a CNN model developed by Google (Khan, Das & Liu, 2024). It is an improved version of the previous Inception models. The Inception V3 model accepts RGB images as input and scales its size to 299 × 299. First, the model uses several convolutional layers to process the input image. Then, the model generates feature maps by adding many Inception modules. Inception modules contain several convolutional layers connected to each other and use parallel pathways. These pathways process features using different filter sizes and depths. Then, these features are combined and their dimensions are reduced to obtain higher-level features. The Inception V3 model also uses techniques such as “batch normalization” and “dropout”. Batch normalization provides normalization of data before each layer and speeds up the training process. Dropout is used to prevent overfitting. Models such as Sufflenet also have features such as channel shuffle, which combines different feature maps to create a better feature map and helps the network learn faster (Zhang et al., 2018). The Inception V3 model creates a classifier with fully connected layers. This classifier is used in tasks such as object recognition, classification and detection.

Proposed deep feature extraction hybrid model

In the first stage of the ALL detection architecture proposed in this study, blood cell images were cropped using the center-based cropping strategy and irrelevant areas were removed from the image. In the second stage, the dataset was divided into train, valid and test, and the test set was kept separate and the Xception architecture was trained. In the next stage, the trained Xception architecture was used as a feature extractor. The fine-tuning and feature extraction steps are given in the visual of the Xception architecture model structure (Srinivasan et al., 2021). The proposed model is presented in detail in Fig. 3. In the last stage of the model, the extracted features were given as input to the XGBoost classifier and the classification results were obtained.

Figure 3 Proposed deep learning model.

Data preprocessing

Preprocessing is the transformation process performed to make raw data more suitable and efficient. When performing classification with deep learning, presenting image data correctly increases the performance of the model. The purpose of preprocessing is to clean the noise, parasites and unnecessary information in the images; to ensure that the model learns correctly and quickly by resizing, normalizing or standardizing. In this study, center-based cropping was applied to segmented blood cell images. The boundaries of the blood cell image were determined using the thresholding method. The center of the blood cell was determined and the image was cropped. The center-based cropping process was performed in two stages.

Determining the center of the image:

(1) Cx=W2,Cy=H2.

Crop points:

(2) x1=Cx−W2+P

(3) y1=Cy−H2+P

(4) x2=Cx+W2−P

(5) y2=Cy+H2−P.

Here, the image width W and height H are shown. P determines the number of pixels to be cropped. The upper left corner of the new cropped image is calculated as (x1,y1) and the lower right corner is calculated as (x2,y2).

Feature extraction

Feature extraction is the process of obtaining meaningful features from the raw data. In this process, features that distinguish the image and represent it with fewer pixels are extracted from the image pixel matrix. The aim of the feature extraction process is to reduce the image size by using less data and to continue the detection process efficiently. Deep learning architectures can be used as feature extractors as well as in tasks such as classification and regression. In this study, pre-trained transfer learning architectures were used to extract features from blood cancer images. Pre-trained CNN architectures such as ResNet, Inception, DenseNet, Xception, VGGNet and MobileNet were used as feature extractors. The Batch Normalization layer was added to the last layer of these architectures and the relevant features were extracted from this layer. In the proposed method, features are extracted using the Xception architecture.

Xception separates the standard convolution operation into a two-step process. These steps are known as Depthwise Convolution and Pointwise Convolution (1 × 1 Convolution). Depthwise Convolution applies convolution to each channel separately. Pointwise Convolution then combines the output channels to enable interaction between them. This operation provides a similar representational power to classical convolutions but with significantly fewer parameters.

The Xception architecture consists of three main blocks: Entry Flow, Middle Flow, and Exit Flow. Each block contains specific building blocks that increase the model’s depth and complexity while maintaining computational efficiency.

Entry Flow is the initial stage of the model and includes the first layers where basic features are extracted from the input image. At this stage, the raw data of the input image is transformed into low-level features. Next come the Middle Flow blocks, which form the core body of the model and consist of multiple repeated layers. The Middle Flow enables higher-level feature representation and contains the layers where the model’s learning capacity is most intense. Finally, the Exit Flow prepares the extracted features for the classification layers; this stage performs the final adjustments necessary for the model’s decision-making process.

Within each block, Depthwise Separable Convolution (DWConv) layers serve as the fundamental building blocks. Unlike traditional convolution, DWConv operates in two stages: first, depthwise convolution is applied separately to each channel, then pointwise convolution with 1 × 1 kernels enables interaction between channels. This structure significantly reduces the number of parameters and computational cost while enhancing the model’s learning capability.

Additionally, the blocks include Batch Normalization layers that help the network learn more stably and efficiently. These layers normalize the output distribution from the previous layer, reducing internal covariate shift and improving the overall training process. The activation function used is the widely adopted rectified linear unit (ReLU), which introduces non-linearity to enable the model to learn complex patterns.

To prevent learning issues such as vanishing gradients as the model depth increases, residual connections (skip connections) are employed between blocks. These connections add the input of a block directly to its output several layers later, facilitating information flow and enabling effective training even in very deep architectures.

The model’s input data has dimensions of (None,224,224,3), where “None” represents a variable batch size. The input images are in three-channel RGB format and are normalized prior to training to ensure faster and more stable learning.

Depthwise separable convolution

One of the fundamental building blocks of the Xception architecture is the depthwise separable convolution (DSC), which performs the conventional convolution operation in two separate steps. This design significantly reduces the computational cost and the number of parameters in the model.

Depthwise convolution: In this step, each channel of the input image is processed independently. That is, for an input with C channels, C separate filters are applied independently. As a result, each channel learns its spatial features without cross-channel interaction.

Mathematically, the ccc-th channel of the input, represented as x(c), is convolved with the corresponding filter k(c) via a 2D convolution operation:

(6) DWConv(x)={x(c)∗k(c)}c=1C.

Here, the ∗ operator denotes the 2D convolution. As a result of this operation, each input channel undergoes spatial filtering independently, producing an output with the same number of channels.

Pointwise convolution: The multi-channel output obtained from the Depthwise Convolution is further processed using 1 × 1 convolutions (pointwise convolutions) to enable interaction and recombination between channels. This step forms linear combinations across channels, generating more complex and richer feature maps.

(7) PWConv(x)=∑c=1C⁡x(c)⋅w(c).

Here, the weight matrix w(c) represents the parameters of the pointwise convolution applied to each channel. This operation enables information transfer and interaction across multiple channels.

Residual connections: As the number of layers in deep neural networks increases, the model faces the problem of vanishing or weakening gradients during training. This issue hinders effective parameter updates and thus impairs learning in deeper layers.

Residual connections were developed to address this problem by directly adding the input of a block to its output several layers ahead. In this way, the model evaluates each block’s output not only with the newly learned features but also with the original information coming from previous layers.

Mathematically, this can be expressed as:

(8) y=F(x)+x.

Here, x denotes the input data to the block, F(x)represents the output resulting from the convolution, normalization, and activation operations applied within the block, and y is the final output of the block after adding the residual connection.

Transfer learning and fine-tuning process

The model is initially initialized with pretrained weights obtained from a large and diverse dataset, such as ImageNet. These pretrained parameters enable the model to recognize fundamental visual features and help accelerate the training process while achieving better generalization performance.

The dataset used for the new task is defined as:

(9) Dnew={(xi,yi)}i=1N.

Here, xi denotes the input image of the i sample in the new task, and yi represents the corresponding expected output (label).

Within the scope of transfer learning, the pretrained weights are further optimized through a fine-tuning process to adapt to the characteristics of the new dataset. During this process, all layers of the model are unfrozen, allowing their weights to be updated and specifically optimized for the new task. This optimization enables the model to learn features unique to the new dataset and enhances classification performance. Particularly, including all layers in training permits the model to adjust previously learned representations to better fit the target data distribution.

The basic steps regarding the moldelin structure are given in Fig. 4.

Figure 4 Architecture of the model.

Classification process

In this study, the features extracted using the transfer learning architecture are classified using machine learning algorithms at this stage. In the proposed method, the features extracted with the proposed Xception model are classified with the advanced extreme gradient boosting (XGBoost) architecture. Extreme Gradient Boosting is a scalable, flexible and fast machine learning algorithm based on the gradient boosting method. It offers high performance especially when working with large data and complex datasets. XGBoost builds models successively using decision trees, and each new model tries to reduce the errors of the previous model. The basic steps of the XGBoost model are explained below.

Objective function

The goal is to minimize both the loss function L and the complexity Ω of the model:

(10) O(t)=L(t)+Ω(t).

Model complexity

The complexity of decision trees is calculated as follows:

(11) Ω(f)=γT+12λ∑j=1T⁡wj2.

Here, T: Number of leaves in the tree, wj represents the leaf values, γ is the penalty per leaf, and λ is the regularization parameter.

(12) L(t)=∑i=1n⁡\Big[gif(xi)+12hif(xi)2\Big]+Ω(f).

In this equation gi=∂L(yi,yi^)∂yi^ symbolizes first derivative and hi=∂L(yi,yi^)∂yi^ symbolizes second derivative.

Optimization of leaf scores

For each leaf j, the weight wj is optimized by the following formula:

(13) wj=−∑i∈Ij⁡gi∑i∈Ij⁡hi+λ.

In this equation Ij set of samples falling on the leaf gi denotes the gradient, and hi denotes the hessian.

Information gain

The benefit of dividing a leaf into two sub-leaves is calculated as follows:

(14) Gain=12[(∑i∈IL⁡gi)2∑i∈IL⁡hi+λ+(∑i∈IR⁡gi)2∑i∈IR⁡hi+λ−(∑i∈I⁡gi)2∑i∈I⁡hi+λ]−γ.

In this formula, IL samples falling on the lower left leaf. IR samples falling on the lower right leaf.

Hyperparameters

In the training phase of the proposed method, all layers of the Xception model trained with ImageNet weights were solved and retrained with the CNMC-2019 dataset. The retrained model was used as a feature extractor. In the training phase, batch normalization, dense(256), dropout and softmax layers were added to the output layer for the performance of the architecture. Transfer learning was applied to the model using the weights previously trained on the ImageNet dataset. Here, some of the parameters in the first layer were frozen and excluded from training. In the meantime, L2 regularization and dropout operations were applied to increase the generalization ability of the model. The hyperparameters used in the training model are given in Table 3. In the XGBoost model employed during the classification phase, the hyperparameters max_depth = 8, n_estimators = 500, and min_child_weight = 2 were selected. The max_depth parameter enables the model to capture deeper structures and complex patterns. In this case, a value of 8 was chosen to balance generalization and learning capacity at an optimal depth. The n_estimators parameter, which defines the number of decision trees constructed during training, serves as a key factor in assessing model performance. The choice of 500 trees was determined based on experimental evaluations conducted on a large dataset, where it yielded optimal results. The min_child_weight parameter represents the minimum sum of instance weights required for a leaf node to undergo further splitting. In this study, a value of 2 was selected to mitigate overfitting while allowing sufficient learning flexibility. A higher value could lead to underfitting, as it would require a greater number of samples per split. Furthermore, L2 Regularization (applied with a value of 0.016) was incorporated to constrain the magnitude of weights, effectively reducing overfitting. Additionally, L1 Regularization was set to 0.006 to promote sparsity within the model. While defining the dropout performance value, 45% of the neurons in the model were randomly deactivated to prevent overfitting. Additionally, a softmax layer was applied for the final prediction, using the ‘categorical_crossentropy’ loss function and the ‘adamax’ optimizer. A learning rate of 1e−2 and max pooling were specified.

Table 3 Hyperparameters of the proposed method.

Hyper parameters	Value	Description	
Input layer size	300, 300, 3	Color	
(Train, validation, test)	70%, 15%, 15%	–	
Weights	ImageNet		
Batch size	32		
Epoch	50	Stopping criteria	
İnitial learning rate	0.001		
Factor	0.5	Valid loss	
Stop patience	10		
Optimizer	Adamax		
Pooling	Max		
BN momentum	0.99		
BN epsilon	0.001		
L2 regularization	0.016		
L1 regularization	0.006		
Dropout	0.45	Dense layer	
Activation	ReLU	Convolution layer	
Classifier	Softmax		
Loss	categorical_crossentropy		

System configuration

NVIDIA GEFORCE RTX 3080 GPU and 32 GB DDR5 RAM were used for training the deep learning architectures used in the experiments performed for ALL detection. OS, OpenCV, Numpy and Pandas libraries were used in the image pre-processing stage. Keras and Tensorflow libraries were used for training the deep learning architectures.

Evaluation metrics

Confusion matrix is one of the basic tools used to evaluate the performance of the model in classification problems. It visualizes the correct and incorrect predictions of the model.

Accuracy: It is the ratio of the number of correctly predicted examples by the model to the total number of examples. It is an appropriate metric if all classes are balanced.

(15) Accuracy=TP+TNTP+TN+FP+FN.

Precision: It shows how many of the samples that the model predicted as positive are actually positive. It is used when false positives are important.

(16) Precision=TPTP+FP.

Recall: It shows how many of the true positive examples were correctly predicted by the model. It is preferred in cases where missed positives are important.

(17) Recall=TPTP+FN.

F-score: It is the harmonic mean that balances between precision and recall. It is a more reliable metric, especially when the data is unbalanced.

(18) F1=2⋅Precision⋅RecallPrecision+Recall.

Results

CNN models are usually composed of a set of variations of specific elements presenting variations in different architectures (Maeda-Gutiérrez et al., 2020). In this study, an evaluation of state of the art pre-trained models for the classification of ALL diseases using images was conducted. In this research, the evaluation of key factors such as accuracy, precision, sensitivity, specificity, F-score and AUC by setting specific hyperparameters and comparison of hybrid CNN models were presented. The performance of the proposed model was compared with hybrid deep learning models and advanced machine learning algorithms.

While applying comparative analyses, the results of hybrid models developed with machine learning algorithms (SVM, KNN, RF, DT, XGBoost) are also given in addition to the performance of each model. The results obtained by all different combinations against the proposed model are presented in detail. Accuracy, precision, recall, F1-score evaluation criteria were taken to evaluate the model performances and the results obtained.

In Table 4, hybrid models were tested with the ResNet50 model and the results are shown.

Table 4 ResNet50 hybrid model results.

Model	Accuracy (%)	Precision (%)	Recall (%)	F1-score (%)	Test time	Per image test time	
SVM + ResNet50	98.06	97.87	97.73	97.80	20.56	0.0126	
RF + ResNet50	98.0	98.22	97.25	97.71	15.11	0.0009	
KNN + ResNet50	98.0	98.06	97.39	97.72	17.10	0.0010	
DT + ResNet50	96.94	96.51	96.55	96.53	15.05	0.0009	
XGBoost + ResNet50	98.38	98.39	97.92	98.15	15.08	0.0009	

When the model performances were examined, the XGBoost + ResNet50 hybrid model gave the highest successful result with an accuracy rate of 98.38%. The F1-score value was determined as 98.15% and it was seen that this result was balanced with precision and sensitivity performances. In addition, the SVM + ResNet50 model stands out as a very successful model with an accuracy rate of 98.06%. The DT + ResNet50 model showed a lower success rate than the other models with an accuracy rate of 96.94%.

The performances of the hybrid models applied together with the Xception model are given in Table 5.

Table 5 Xception hybrid model results.

Model	Accuracy (%)	Precision (%)	Recall (%)	F1-score (%)	Test time	Per image test time	
SVM + Xception	98.88	98.87	98.58	98.72	19.50	0.0121	
RF + Xception	98.56	98.44	98.30	98.37	14.40	0.0009	
KNN + Xception	98.81	98.77	98.53	98.65	15.40	0.0010	
DT + Xception	97.62	97.35	97.26	97.31	14.38	0.0009	
XGBoost + Xception	98.88	98.82	98.63	98.72	14.35	0.0009	

SVM + Xception reached the highest sensitivity value with 98.58% and XGBoost + Xception reached 98.63%. In addition, both models showed the highest success with 98.88% accuracy. This situation reflects that the number of correctly classified examples is quite high in the total model sample. When the performance of these models is examined, it is seen that the false positive rates are low as well as the correct prediction rates. On the other hand, the DT + Xception model has the lowest accuracy value with 97.62%. Since DT + Xception generally shows lower performance, it may not be preferred for more complex problems.

The performances of the hybrid models integrated into VGGNet-19 are shown in Table 6.

Table 6 VGG19 hybrid model results.

Model	Accuracy (%)	Precision (%)	Recall (%)	F1-score (%)	Test time	Per image test time	
SVM + VGG19	93.00	92.70	91.29	91.94	36.36	0.0227	
RF + VGG19	92.00	92.10	89.58	90.67	31.27	0.0195	
KNN + VGG19	91.81	91.33	89.82	90.57	32.26	0.0201	
DT + VGG19	89.75	88.44	88.29	88.37	31.21	0.0195	
XGBoost + VGG19	93.31	92.91	91.82	92.33	31.25	0.0195	

As seen in Table 6, all models exhibited high classification performance with VGG19 based feature extraction. The highest accuracy and F1-score values were obtained with the XGBoost + VGG19 model (93.31% accuracy, 92.33% F1-score). The SVM + VGG19 model also showed similarly high performance and stands out with its 91.94% F1-score value. The other models (RF, KNN and DT), although algorithms have relatively lower performance, are at an acceptable level. Especially the DT + VGG19 model has the lowest metric values compared to the others, which shows that decision trees provide limited success with complex features based on deep learning.

Accordingly, the performance results of the EfficientNetB3 model created are presented in detail in Table 7.

Table 7 EfficientNetB3 hybrid model results.

Model	Accuracy (%)	Precision (%)	Recall (%)	F1-score (%)	Test time	Per image test time	
SVM + EfficientNetB3	98.44	98.35	98.11	98.23	20.37	0.0127	
RF + EfficientNetB3	98.75	98.68	98.49	98.58	15.28	0.0095	
KNN + EfficientNetB3	98.62	98.68	98.20	98.43	16.27	0.0196	
DT + EfficientNetB3	98.12	97.75	98.02	97.88	15.29	0.0095	
XGBoost EfficientNetB3	98.69	98.49	98.54	98.51	15.25	0.0095	

As a result of the classification of deep features extracted using EfficientNetB3 architecture with different machine learning algorithms, it is seen that all models achieve very high success rates. The highest accuracy rate was obtained in the random forest (RF + EfficientNetB3) model with 98.75%. This is followed by XGBoost, KNN, and SVM models, respectively. In particular, the RF, KNN, and XGBoost models produced very close and balanced results in terms of both accuracy and precision, recall, and F1-score. The SVM + EfficientNetB3 model exhibited a remarkable performance with 98.44% accuracy and 98.23% F1-score.

The performance results of the hybrid models developed with DenseNet-121 are shown in Table 8.

Table 8 DenseNet121 hybrid model results.

Model	Accuracy (%)	Precision (%)	Recall (%)	F1-score (%)	Test time	Per image test time	
SVM + DenseNet121	98.56	98.35	98.40	98.37	21.50	0.0134	
RF + DenseNet121	98.00	97.78	97.69	97.73	16.41	0.0102	
KNN + DenseNet121	98.44	98.35	98.11	98.23	17.40	0.0162	
DT + DenseNet121	97.19	96.75	96.89	96.82	16.35	0.0102	
XGBoost + DenseNet121	98.31	98.11	98.06	98.09	16.38	0.0102	

As a result of the classification of the features obtained with the DenseNet121 architecture with different machine learning algorithms, high accuracy rates were achieved. The most successful model was SVM + DenseNet121 with 98.56% accuracy and 98.37% F1-score. This model stands out by exhibiting a balanced and strong performance in all metrics. The KNN + DenseNet121 and XGBoost + DenseNet121 models also produced very close results with 98.44% and 98.31% accuracy rates, respectively, and maintained a high precision-recall balance.

In MobileNetv2, calculations were performed using one depthwise convolution and one pointwise convolution, and the results obtained are given in Table 9.

Table 9 MobileNetV2 hybrid model results.

Model	Accuracy (%)	Precision (%)	Recall (%)	F1-score (%)	Test time	Per image test time	
SVM + MobileNetV2	97.88	97.92	97.25	97.58	15.23	0.0094	
RF + MobileNetV2	98.00	97.92	97.54	97.73	10.14	0.0062	
KNN + MobileNetV2	98.19	98.25	97.63	97.93	11.13	0.0068	
DT + MobileNetV2	97.12	96.62	96.89	96.75	10.08	0.0062	
XGBoost + MobileNetV2	98.19	98.06	97.83	97.94	10.11	0.0062	

When Table 9 is examined, the highest accuracy rate of 98.19% was observed in both KNN + MobileNetV2 and XGBoost + MobileNetV2 models. These models also attracted attention in terms of F1-score with values of 97.93% and 97.94%, respectively. RF + MobileNetV2 and SVM + MobileNetV2 models also produced very close results and showed a success rate of over 97.5%.

The hybrid models integrated with InceptionResNetv2 and the performance results are shown in Table 10.

Table 10 InceptionResNetV2 hybrid model results.

Model	Accuracy (%)	Precision (%)	Recall (%)	F1-score (%)	Test time	Per image test time	
SVM + InceptionResnetV2	98.38	98.30	98.01	98.15	28.69	0.0179	
RF + InceptionResnetV2	98.19	98.11	97.78	97.94	23.60	0.0147	
KNN + InceptionResnetV2	98.12	97.97	97.78	97.87	24.60	0.0153	
DT + InceptionResnetV2	97.06	96.42	96.99	96.69	23.54	0.0147	
XGBoost + InceptionResnetV2	98.31	98.11	98.06	98.09	23.57	0.0147	

The classifications made with the InceptionResNetV2 architecture attract attention with high accuracy rates and well-balanced performance metrics in all models. The highest accuracy rate was obtained in the SVM + InceptionResNetV2 model (98.38%), and it was observed that this model also showed high success in all metrics with an F1-score value of 98.15%. This indicates that the SVM model works in strong harmony with InceptionResNetV2 and makes accurate classifications.

The results of the Inceptionv3 trained on the ImageNet dataset are shown in Table 11.

Table 11 InceptionV3 hybrid model results.

Model	Accuracy (%)	Precision (%)	Recall (%)	F1-score (%)	Test time	Per image test time	
SVM + InceptionV3	98.69	98.68	98.34	98.51	20.46	0.0127	
RF + InceptionV3	98.31	98.20	97.97	98.08	15.38	0.0095	
KNN + InceptionV3	98.44	98.39	98.06	98.22	16.40	0.0101	
DT + InceptionV3	97.44	97.25	96.93	97.09	15.31	0.0095	
XGBoost + InceptionV3	98.56	98.35	98.40	98.37	15.35	0.0095	

The SVM + InceptionV3 model stands out with 98.69% accuracy, and with 98.51% F1-score, it is understood that the model shows a strong performance in all metrics. It can be said that SVM is quite successful in matching the features extracted by InceptionV3 and is effective in general classification. Another remarkable model is XGBoost + InceptionV3. With 98.56% accuracy rate and 98.37% F1-score, this model provided a stable performance and reached high precision and recall values. This shows that XGBoost works very efficiently with deep learning features.

Proposed hybrid model results

In this hybrid model, Xception was used to extract complex features, while XGBoost was used to perform the classification steps using these features. With the parameter settings developed in accordance with the model, the meaningful features obtained from the Xception model were given as input to the XGBoost model with the feature vector in the last layer of the model. The classification process was performed with the peer tree model created via XGBoost. The data on the classification performance of the model are shown in Table 12.

Table 12 Hybrid model classification report.

Class	Precison	Recall	F1-score	Support	
0	98.66	97.91	98.28	526	
1	98.98	99.35	99.16	1,074	
Macro avg	98.82	98.63	98.72	1,600	
Weighted avg	98.87	98.88	98.87	1,600	

Table 13 shows the comparative analysis table of the hybrid model with other models. When these results are examined, it is seen that the developed Xception + XGBoost model achieves higher accuracy than other deep models with 98.88% accuracy. Similarly, when the results of each CNN model are examined with hybrid machine learning algorithms, it is seen that the proposed model reaches the best accuracy value.

Table 13 Comparative analysis of the proposed model with deep learning models.

Model	Accuracy (%)	Precision (%)	Recall (%)	F1-score (%)	AUC (%)	Total parameter	
ResNet50	95.06	95.26	93.29	94.19	97.0	24.120.962	
Xception	95.94	95.76	94.82	95.27	97.3	21.394.730	
VGG19	92.31	92.79	89.33	90.79	95.0	20.158.274	
EfficientNetB3	95.87	95.87	94.56	95.18	99.0	11.183.665	
DenseNet-121	95.69	95.37	94.64	94.99	97.0	7.304.514	
MobileNetV2	94.06	93.65	92.55	93.07	98.0	2.591.554	
InceptionResNetV2	96.19	95.95	95.21	95.57	98.0	54.736.866	
InceptionV3	95.37	95.17	94.09	94.61	98.0	22.336.034	
Proposed model	98.88	98.82	98.63	98.72	99.0	17.246.551	

The proposed model demonstrates robust performance, achieving an F1-score of 98.72% and an AUC of 99%. While the Xception model attains high accuracy (95.94%) and an F1-score of 95.27%, its parameter complexity remains at a moderate level. Similarly, ResNet50 delivers strong performance with an accuracy of 95.94%; however, it falls short compared to other state-of-the-art models. The confusion matrix and ROC curve results for the proposed model are illustrated in Fig. 5.

Figure 5 Hybrid model confusion matrix (A) and ROC (B).

A heat map comparing the performance metrics of different models is given in Fig. 6. The heat map represents the values of each model in various performance metrics with color tones. The color tones vary according to the magnitude of the metric values, with higher values being shown in darker or more vibrant colors. The heat map allows for a quick comparison of the performance of the models and identifies which model performs better in which metric. Especially when comparing multiple metrics, it is easy to understand which model is better overall thanks to the different color tones. The values of these metrics are given here for each model.

Figure 6 A heat map comparing performance metrics of different models.

The radar chart comparing the performance metrics of different models is given in Fig. 7. This type of chart is used to visualize multiple variables (metrics) at the same time, and performance comparisons are made especially between different categories or models. The radar chart is drawn on axes originating from a center point, and each axis is planned to represent a different variable. This chart determines which areas a model or category is strong in and which areas it is weak in. When the figure is examined, AUC (blue line) is generally one of the metrics with the highest values. Although there are small differences between precision (red) and recall (green), they generally show a similar distribution. F1-score (purple) and accuracy (blue) show consistently high performance.

Figure 7 Radar chart of performance metrics of various deep learning models.

When Fig. 8 is examined, it is seen that the VGG-19 recall value is lower than the other models. When compared to the studies in the literature, it is seen that the hybrid model has higher performance. At the same time, it is seen that this model has higher performance compared to the studies done with ensemble models.

Figure 8 Performance indicators of hybrid model.

To reduce bias and ensure a more reliable assessment of the classification model’s performance, we employed the 10-Fold Cross-Validation technique on the dataset used in this study. In this approach, the dataset is randomly partitioned into 10 equal subsets (folds). In each iteration, one fold is reserved as the validation set while the remaining nine folds are used for training. This process is repeated 10 times so that each subset is used exactly once as the validation data. The final performance metrics are then averaged over all folds, providing a robust estimation of the model’s generalization ability.

This method is particularly effective in reducing overfitting risks and performance variance due to random train-test splits, especially when working with limited or imbalanced datasets. By using all data points in both training and validation at different iterations, 10-fold cross-validation offers a statistically sound framework to evaluate classification models.

To quantify the consistency of the results, the standard deviation (Std) of the accuracy values obtained across folds was also calculated. This helps in assessing the stability of the model’s performance across different subsets of the data.

Table 14 presents the accuracy scores obtained from each fold, along with the computed standard deviation, based on the application of the 10-fold cross-validation technique on the dataset used in this study.

Table 14 Performance evaluation of the proposed model based on 10-fold cross-validation.

Fold	Accuracy	Precision	Recall	F1-score	AUC	Std	
Fold 1	0.9821	0.9797	0.9945	0.9870	0.9936	0.0024	
Fold 2	0.9849	0.9850	0.9931	0.9890	0.9987	0.0024	
Fold 3	0.9840	0.9863	0.9903	0.9883	0.9983	0.0024	
Fold 4	0.9803	0.9809	0.9903	0.9856	0.9948	0.0024	
Fold 5	0.9849	0.9863	0.9917	0.9890	0.9960	0.0024	
Fold 6	0.9887	0.9917	0.9917	0.9917	0.9984	0.0024	
Fold 7	0.9849	0.9889	0.9889	0.9889	0.9948	0.0024	
Fold 8	0.9887	0.9917	0.9917	0.9917	0.9986	0.0024	
Fold 9	0.9859	0.9810	0.9986	0.9897	0.9972	0.0024	
Fold 10	0.9849	0.9889	0.9889	0.9889	0.9972	0.0024	

The proposed model has achieved remarkably high and consistent performance metrics during the 10-fold cross-validation process. In particular, the high AUC and F1-scores demonstrate that the model is a reliable classifier in terms of both accuracy and balance. The low standard deviation indicates that the model performs similarly well across different subsets of the data, supporting its robustness. These results suggest that the model can be effectively utilized in real-world applications.

To further evaluate the classification performance of the model on each fold, receiver operating characteristic (ROC) curves were plotted separately for each fold during the 10-fold cross-validation process. The ROC curve is a widely used tool that illustrates how well a model can distinguish the positive class, especially in binary classification tasks, by showing the trade-off between sensitivity (true positive rate) and specificity (false positive rate).

Each ROC curve was generated based on the decision thresholds of the model’s predictions for the corresponding fold. These curves allow for a visual assessment of the model’s discriminative ability across different subsets of the data. The similarity of the curves indicates the consistency and stability of the model’s performance.

Figure 9 presents the ROC curves obtained for each fold in the 10-fold cross-validation. This visual representation provides a more concrete understanding of the model’s sensitivity and generalizability across different partitions of the dataset, thereby strengthening the credibility of the study.

Figure 9 ROC curves for each fold of the proposed model.

Discussion

This section investigates the results of the proposed approach in detail from different aspects. Also, the accuracy achieved by the proposed approach is compared with the state-of-the-art methods as shown in the literature review. In the study conducted with classical deep learning models VGG16 and VGG19, it is seen that lower accuracy is achieved with 91.59% compared to more modern models (Maurício de Oliveira & Dantas, 2021). In this study, the VGG19 algorithm achieved a higher accuracy of 92.31% with the hyperparameter changes applied. Despite the fact that DenseNet-121 has a lower number of parameters and a longer training time, it fell behind other architectures despite the 86.15% accuracy it achieved in a study (Duggal et al., 2017). It is seen that the method presented in this study reaches a high performance value of 95.69% in the DenseNet-121 model. Similarly, in another study where augmentation, rebalancing and ROI extraction methods were applied for data preprocessing, 82.9% accuracy was obtained (Mondal et al., 2021). This study supports studies that can be improved with hybrid models, using deep learning (Xception) for feature extraction and XGBoost or SVM for classification. The success of such advanced methods shows that these techniques can make more accurate predictions with the remaining data structures after removing the outliers. The fact that hybrid models achieve higher accuracy with Xgboost and the classification performance of F1-scores also support the preference of the algorithm. The proposed model not only outperformed the existing models such as ResNet-50, VGG19, DenseNet121 and EfficientNet in the literature; it also provided a more balanced, interpretable and applicable model structure. Especially, the Xception + XGBoost hybrid successfully provides the balance of high accuracy and low computational cost by combining deep feature learning with a classical robust classifier. The proposed model is superior to most of the existing state-of-the-art (SOTA) models not only in terms of classification accuracy but also in terms of the number of parameters and hence computational efficiency. In this respect, the applicability of the model in low-hardware environments increases and its clinical validity is strengthened. Especially against models such as InceptionResNetV2, which offer high accuracy but contain very high parameters, the high accuracy-low complexity balance offered by the proposed structure is a tangible contribution to the literature. The proposed model exhibited the highest overall performance with 98.88% accuracy, 98.82% precision, 98.63% recall, 98.72% F1-score and 99% AUC. This draws attention not only with its high correct classification rates but also with its successful establishment of balance between classes. Among the other models compared, InceptionResNetV2 ranks second with 96.19% accuracy, while it has a considerably high number of parameters. The proposed model provides this accuracy with only 17.2 million parameters at a significantly lower computational cost. This hybrid structure, which is obtained by combining the powerful feature extraction capacity of the Xception architecture with the high classification performance of XGBoost, offers a more explainable and flexible approach in contrast to classical CNN-based models. When similar studies in the literature are examined, it is observed that most models are either very high parameterized (e.g., InceptionResNetV2) or low parameterized but weak in terms of accuracy (e.g., MobileNetV2). The proposed model proposes a new optimization line by establishing a balance between these two extremes, offering lower hardware requirements with high accuracy. The proposed model offers higher accuracy with 68% fewer parameters than InceptionResNetV2. This shows that the model offers a great advantage in terms of computational efficiency and clinical applicability. Older architectures such as VGG19 have fallen behind current models in terms of both accuracy and efficiency. In the study utilizing the Bayesian optimization algorithm, the proposed ensemble model achieved an accuracy of 96.26% (Huang & Huang, 2024). During the training phase, classical models such as InceptionV3, EfficientNetB4, and ResNet50 were employed. In this context, ResNet50 yielded an accuracy of 82.83%, while InceptionV3 achieved 84.82%. However, in the proposed study, both models demonstrated significantly higher performance, with ResNet50 reaching 95.06% accuracy and InceptionV3 achieving 95.37%, aligning with the overall superior performance of the model. This finding is also consistent with trends observed in the recent literature.

Simultaneously, it is seen that the hyperparameters presented in the study increase the performance in the normal models of the algorithms as well as in the hybrid models. In addition, it is seen that the preprocessing steps increase the performance of the model. In the application of the multiple learning model that collects global and local features in a single area, 91.89% F1-score was obtained (Mathur et al., 2020). Although cropping, attention reduction, and hyperparameter methods were applied in the study, the results obtained fell behind this study. This finding supports the hypothesis that the observed differences are not due to random feature selection, but also that the selected method affects the performance.

The highest F1-score was obtained in the study when the learning rate was 0.0001. In contrast, the highest accuracy was achieved in this study when the learning rate was 0.001 and L1 regularization was 0.006. In another study published on the ALL dataset, 95.22% recall and 93.35% precision values were achieved (Mei et al., 2025). Maurício de Oliveira & Dantas (2021) proposed simple changes for high-performance malignant leukocyte classification in their study. VGG16, VGG19 and Xception architectures are used in their study. The dataset was expanded and more diversity was achieved by using data augmentation methods to balance the training and validation sets. Different image processing transformations were used to produce the augmented images. At the end of the study, the performance of the algorithms trained on the dataset is VGG16: 92.48%, VGG19: 91.59%, Xception 90.41%, respectively. As an older model, VGG19 has lower metrics in the proposed model with 92.31%. It is seen that the hybrid model reaches the highest accuracy, precision, recognition and F1-score values compared to CNN algorithms. When we analyze each algorithm, it is seen that the InceptionResNetV2 model achieves the highest accuracy after the proposed model. When the performance of the model is compared with the results of hybrid models, it is seen that higher accuracy values are achieved. When the performances of the models are examined comparatively, although the XGBoost + ResNet50 model is the highest hybrid model with 98.69% accuracy, it has lower results than the proposed model. The XGBoost + InceptionV3 hybrid model is another hybrid model that comes after the proposed model with 98.56% accuracy. When the hybrid models made in the literature are examined, the NASNetLarge + VGG19 transfer learning model has an accuracy value of 96.58% (Kasani, Park & Jang, 2020). In the study, similar preprocessing and data augmentation methods were applied in the test processes. However, it is observed that the learning parameters and computational requirements utilized in the study are considerably high.

In another study proposing a hybrid model, an accuracy of 98.5% was achieved (Duggal et al., 2017). Liu et al. (2022) developed a deep learning model named Advanced Imaging and Microscopy Integrated Classification (AIMIC) for the classification of microscopic images using deep learning techniques. This model is used to automatically classify microscopic images using deep neural networks, CNN’s and long-short term memory (LSTM). In this study, four different datasets are used, namely C-NMC, ALL-IDB2, PBC and LISC. Among the models, the highest accuracy rate of 96.93% was obtained by ResNeXt-50-32×4d. This model was obtained by changing the structure of ResNet-52.

In another study (Kaur & Singh, 2024), a new hybrid deep neural network was presented for accurate and timely detection of Leukemia. Here, VGG16-PCA-PB3C achieved 95% accuracy and 94.0% sensitivity for leukemia detection. At the same time, the processing times and hyperparameter optimizations including the obtained results were not given in detail in the study. In a hybrid study (Shehta, Nasr & El Ghazali, 2025) conducted with different ResNet models, the performance of ResNetRS50 and RegNetX016 models in cancer detection was evaluated. In this study, an accuracy rate of 97% was achieved with ResNetRS50. When compared with current ResNet models, it is also seen that the performance of the proposed model reaches higher accuracy in the CNMC dataset compared to different ResNet models. On the contrary, the obtained results are lower than the models presented in this study.

In our study, the ResNet50 model reached a high accuracy value of 95.06% with its original structure. In addition, among the hybrid models developed with the ResNet50 model, the XGBoost + ResNet50 combination exhibited a very high performance with an accuracy rate of 98.38%. This result is another indication that the backbone structure of the ResNet50 model has a strong feature extraction ability.

Conclusion

The results obtained in this study show that the deep learning-based hybrid model provides a higher accuracy of 98.69% compared to traditional methods. The hybrid use of Xception and XGBoost brings together the strengths of deep learning and machine learning. It has been observed that there is an improvement in accuracy by training deep learning architectures with correctly optimized hyperparameters with different CNN structures. To increase the model performance, image preprocessing stages were customized using different methods such as feature selection, data augmentation, and data thresholding. The optimal threshold value in the image was determined, the pixel value in the image was compared with the threshold value T, and meaningful features were extracted from the image. There are some limitations of this study that should be addressed in future studies.

Firstly, this study presents hybrid models for the detection of blood cells. It also estimates blood cells according to two different input parameters. However, there may be other classifiers such as classification clusters belonging to different formations of blood cells. In addition, apart from the feature extraction process, features obtained from the person’s blood values with manual features can be given as input to the algorithms as parameters.

In future studies, in addition to blood cell estimation, basic factors such as counting blood cells and possible metastasis can be added to the studies. Although the classification process was carried out with hybrid models in this study, scenario-based models that include different assumptions, strategies and doctor opinions can be developed.

At the end of the study, it was seen that while the Xception-based model extracts complex features from images, the XGBoost classifier can effectively use these features in classification or regression applications. This approach can be shown as a powerful solution especially for processing complex image data and making fast predictions. This is an important strategic development for new models and approaches to be developed.

Limitations

The generalizability of the model is affected by several limitations. First, the different standards of the images used may cause the features learned by the model during training to be inconsistent across different environments and devices. This may negatively affect the accuracy of the model unless it is tested on real-world data. In addition, the high GPU requirements of deep learning models may limit their widespread use in clinical settings, as such hardware may not always be available in all environments. When the model is trained on small and imbalanced datasets, it carries the risk of overfitting, which may cause the model to perform poorly on a larger dataset by only fitting the training data. Finally, the use of deep learning models in clinical settings may pose challenges in terms of security, interpretability, and compatibility.

Supplemental Information

Supplemental Information 1 Code.

Supplemental Information 2 Introduction, explanation and implementation steps of the code.

Additional Information and Declarations

Competing Interests

The authors declare that they have no competing interests.

Author Contributions

Gökalp Çınarer conceived and designed the experiments, performed the experiments, analyzed the data, performed the computation work, prepared figures and/or tables, authored or reviewed drafts of the article, and approved the final draft.

Data Availability

The following information was supplied regarding data availability:

The dataset is available at Mourya, S., Kant, S., Kumar, P., Gupta, A., & Gupta, R. (2019). ALL Challenge dataset of ISBI 2019 (C-NMC 2019) (Version 1) [dataset]. The Cancer Imaging Archive. https://doi.org/10.7937/tcia.2019.dc64i46r.

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
