# Peer review of "Hybrid deep layered network model based on multi-scale feature extraction and deep feature optimization for acute lymphoblastic leukemia anomaly detection"

_PeerJ Computer Science, doi:10.7717/peerj-cs.3174_

## Round 0.1 · original submission · Major Revisions

Dear authors,

You are advised to critically respond to all comments point by point when preparing an updated version of the manuscript and while preparing for the rebuttal letter. Please address all comments/suggestions provided by reviewers, considering that these should be added to the new version of the manuscript.

Kind regards,
PCoelho

Reviewer 1 ·

Basic reporting

The work entitled “Hybrid deep-layered network model based on multi-scale feature extraction and deep feature optimization for acute lymphoblastic leukemia anomaly detection (#115747) was thoroughly examined. The following points were observed.

The authors deserve appreciation for their efforts, which provide high-quality observational statistics. They are asked to incorporate the following comments from the reviewer.

1. Clear and unambiguous, professional English used throughout.

2. 2018 data is provided in the Introduction, including statistics on leukemia for 2024 or 2025.

3. Don't use words like "they" in literature reviews.

Experimental design

1. Give the dataset details in Tabulation format, not as a paragraph format
2. No need for figure 4
3. Figures 2 and 3 are not clearly visible

Validity of the findings

1. Observed results are already tabulated. Give the inference in detail in the discussion section. The tabulated data are repeated as paragraphs
2. What do you mean by Traditional methods here?
3. Tabulate the Computational time for each hybrid model
4. Cropping is done as a segmentation process. Instead, the authors can use any segmentation algorithm or deep learning model like UNet.
5. Use glyph plots, pie, and bar charts instead of a larger number of tabulations

Reviewer 2 ·

Basic reporting

While the paper is generally well structured and addresses an important topic, the description of the proposed methodology lacks sufficient depth and clarity. To strengthen the overall quality of the manuscript, it is strongly recommended that the authors revise.

1. The methodology section should be more thoroughly articulate each stage of the proposed approach.

2. Specifically, the inclusion of a detailed workflow diagram, a comprehensive depiction of the model architecture,

3. Precise mathematical formulations, particularly feature extraction and classification processes, would significantly improve the transparency, reproducibility, and scientific robustness of the study.

4. The authors have included several mathematical equations; however, detailed explanations of the variables used are lacking. It would be beneficial if the authors added a nomenclature section that lists all variables along with their respective definitions and explanations. Additionally, the mathematical formulation of the feature extraction and classification processes should be presented clearly, accompanied by comprehensive explanations to enhance the reader’s understanding of the underlying methodology.

Experimental design

The authors have explained in detail the relationship to the C-NMC-2019 dataset that has been used to experiment (training, validation, and testing). However, the paper lacks focus and systematic presentation in describing both the results and the experimental methods employed. The experimental design is insufficiently detailed and should be organized in a clear and comprehensive manner to enable readers to fully understand how the model evaluation was conducted. Specifically, the paper does not explicitly clarify the experimental setup, such as whether the model was tested using cross-validation methods (e.g., K-Fold Cross-Validation), hold-out validation, or Monte Carlo approaches. Providing this information is crucial for assessing the validity and generalizability of the model.

Furthermore, the authors do not provide detailed explanations regarding the hyperparameter settings used for each model, particularly for the Xception and XGBoost components within the proposed hybrid framework. Essential details such as learning rate, number of epochs, batch size, tree depth, number of estimators, and regularization parameters for XGBoost should be reported to ensure experiment reproducibility and to evaluate the impact of hyperparameter tuning on model performance.

Validity of the findings

This study introduces a hybrid model combining Xception and XGBoost that achieves high accuracy for leukemia (ALL) classification. The authors have taken important steps, such as optimizing image preprocessing and feature extraction to enhance model performance. However, the explanation of the novelty and impact of this research compared to existing studies remains unclear, making it difficult to fully understand its specific contributions.

The conclusions are consistent with the reported results, and the authors have acknowledged several limitations, including variations in image quality, high hardware requirements, small dataset size, and challenges in clinical deployment. However, crucial details regarding the experimental methodology, such as the model validation approach (e.g., whether K-Fold Cross-Validation or other methods were used) and the hyperparameter settings (e.g., learning rate, number of epochs) are not provided. Including this information is essential for a clear understanding of the study and for enabling meaningful replication by other researchers.

Additional comments

This paper addresses an important topic with a promising hybrid approach in the field of medical image classification. However, the authors should provide a more detailed and structured explanation of the experimental procedures, including how model validation was conducted and how hyperparameters were set. This information is crucial to enable others to accurately replicate the study and assess the robustness of the model.

Additionally, the authors need to discuss the practical implications more thoroughly, such as the high hardware requirements and how these limitations might be addressed in clinical applications. A clearer comparison between this hybrid approach and other state-of-the-art methods, particularly regarding novelty and advantages, would greatly help readers understand the specific contributions of the work.

It is recommended to show the experimental results in graphs for each epoch, so we can see how the model improves during training. The authors should also clearly explain if the reported accuracy is the average accuracy, the highest accuracy, or something else. Additionally, including ROC curves and AUC values for each result is important to better evaluate the model’s performance

---

## Round 0.2 · Minor Revisions

Dear authors,

Thanks a lot for your efforts to improve the manuscript.

Nevertheless, some concerns are still remaining that need to be addressed.

Like before, you are advised to critically respond to the remaining comments point by point when preparing a new version of the manuscript and while preparing for the rebuttal letter.

Kind regards,
PCoelho

Reviewer 1 ·

Basic reporting

Good

Experimental design

Good

Validity of the findings

Good

Additional comments

The revised manuscript is well and good in all aspects

Reviewer 2 ·

Basic reporting

The authors have made several revisions; however, there is still no diagram that illustrates the model architecture and the overall system workflow. It is recommended that the authors include a diagram that clearly explains each step of the process, from data input to the performance evaluation stage. Please provide a complete illustration of the model workflow and architecture, along with a clear explanation of each component, to help readers gain a comprehensive understanding of the proposed method.

Experimental design

Although the authors have provided explanations regarding the dataset and the general flow of experimentation (training, validation, and testing), the manuscript still does not clearly specify the experimental model or validation strategy employed. It remains unclear whether the authors used K-Fold Cross-Validation, hold-out validation, Monte Carlo validation, or another evaluation method. Therefore, the authors are strongly encouraged to explicitly state, demonstrate, and explain the experimental scenario and validation method used, along with any relevant details.

Validity of the findings

The study presents promising results with the proposed hybrid model combining Xception and XGBoost for leukemia (ALL) classification. However, the findings do not clearly specify the experimental validation model employed. It remains unclear which experimental approach was used to evaluate the model’s performance, such as K-Fold Cross-Validation, hold-out validation, or other methods. Therefore, the authors are strongly encouraged to clearly state, describe, and complete the details of the experimental validation framework applied in this study.

Additional comments

The authors have presented the experimental results in graphs for each epoch, allowing us to see how the model improves during training. However, the ROC curves and AUC values provided represent only a single experimental result, as shown in Figure 5. In my opinion, this revision still does not meet the publication standards. Therefore, a comprehensive revision is necessary before the manuscript can be considered for publication.

---

## Round 0.3 · accepted · Accept

Dear authors, we are pleased to verify that you meet the reviewer's valuable feedback to improve your research.

Thank you for considering PeerJ Computer Science and submitting your work.

Kind regards
PCoelho

Reviewer 2 ·

Basic reporting

The author has made revisions in accordance with the review results.

Experimental design

Experimental results have been shown in this paper and the authors have explained comprehensively

Validity of the findings

The author has completed all sections in accordance with the minimum standards of the journal.

Additional comments

The paper can be published in PeerJ after proofreading has been completed properly